# Post-Mortem Interval Estimation Based on Insect Evidence: Current Challenges

**DOI:** 10.3390/insects12040314

**Published:** 2021-04-01

**Authors:** Szymon Matuszewski

**Affiliations:** 1Laboratory of Criminalistics, Adam Mickiewicz University, Święty Marcin 90, 61-809 Poznań, Poland; szymmat@amu.edu.pl; 2Wielkopolska Centre for Advanced Technologies, Adam Mickiewicz University, Uniwersytetu Poznańskiego 10, 61-614 Poznań, Poland

**Keywords:** forensic entomology, carrion insects, development, succession, validation

## Abstract

**Simple Summary:**

The post-mortem interval of human cadavers may be estimated based on insect evidence. In order to identify scientific challenges that pertain to these estimations, I review forensic entomology literature and conclude that research on the development and succession of carrion insects, thermogenesis on cadavers and the accuracy of PMI estimates are of primary importance to advance this field.

**Abstract:**

During death investigations insects are used mostly to estimate the post-mortem interval (PMI). These estimates are only as good as they are close to the true PMI. Therefore, the major challenge for forensic entomology is to reduce the estimation inaccuracy. Here, I review literature in this field to identify research areas that may contribute to the increase in the accuracy of PMI estimation. I conclude that research on the development and succession of carrion insects, thermogenesis in aggregations of their larvae and error rates of the PMI estimation protocols should be prioritized. Challenges of educational and promotional nature are discussed as well, particularly in relation to the collection of insect evidence.

## 1. Introduction

Carrion insects living in human cadavers can be highly useful for the estimation of the post-mortem interval (PMI) [1,2]. Methods for PMI estimation based on insect evidence are developed, validated, improved and applied by forensic entomologists. This field is growing with a constant increase in the number of scientific publications and countries where entomology-based estimation of PMI is regularly used in death investigations [3,4]. As a maturing field, forensic entomology contains several weaknesses and under-researched areas. These challenges are the focus of this article.

A PMI estimate is only as good as it is close to the true PMI. The accuracy of estimation is most important, particularly for the end users of insect evidence. Therefore, the major general challenge for the field is to reduce the estimation inaccuracy. Its sources are related to both the collection and analysis of insect evidence (Figure 1). I divided this paper into sections devoted to the collection of insect evidence, research on insect development and succession, reconstructing temperature conditions, analysis of challenging evidence and validation of the protocols for PMI estimation.

## 2. Collection of Insect Evidence

Errors in the collection of insect evidence are certainly among the most important sources of the inaccuracy in PMI estimation. Death scene samples frequently misrepresent cadaver entomofauna. However, it is difficult to discern how bad these samples usually are and what the consequences of sampling errors are for the estimation of PMI. In most cases insects are collected by law enforcement officers or medical examiners, and rarely by entomologists. In a recent case, insects were sampled by police officers with the medical examiner and independently by entomologists, which enabled—in this paper—the comparison of samples taken by non-experts and experts [5]. The sample taken by non-experts was distinctly less diverse and did not contain insect evidence, based on which PMI has finally been estimated (see Table 1 and Table 2 in [5]). If PMI was estimated in this case using only the non-expert sample, no meaningful maximum PMI would be derived, although the minimum PMI would be similar to the one estimated based on the expert sample (unpublished data). Another kind of error in the collection of evidence is the error of preservation. Insects may be preserved improperly, for example using an unsuitable preservative or a leaking container [6,7]. Such errors may limit the scope of possible analyses and in extreme cases may even destroy the evidence. Although there are no surveys of errors in the collection of insect evidence, I think that most experts share the opinion that insect samples frequently misrepresent cadaver entomofauna or are preserved improperly. We should therefore discuss whether our guidelines for the collection of evidence are truly fit-for-purpose.

Guidelines for the collection of insect evidence state that death scene samples should accurately represent cadaver entomofauna, i.e., all life stages of each important species that inhabit a cadaver should be represented in the sample [8,9,10]. Cadaver entomofauna may be very diverse and abundant, consisting of many life stages from many species, some in very large numbers. However, to estimate PMI only a small part of it is necessary. Usually, we choose the most developmentally advanced life stage of the most successionally advanced species, and even if the PMI estimate is based on a larger number of taxa, this is usually no more than two or three [11]. Therefore, in most cases a representative sample is redundant, and for this reason we should reshape guidelines for the collection of insect evidence and abandon our commitment to the true representativeness of death scene samples. I believe it is possible to develop guidelines that are user-friendly, quick to implement and that yield more fit-for-purpose samples, i.e., the most developmentally and successionally advanced insects only (Table 1). Insects are usually collected by the law enforcement officers with basic skills in entomology, whereas guidelines for the collection of insect evidence are usually addressed to entomologists. Therefore, we should provide guidelines for non-entomologists that specify what insect evidence they should look for and where it can be found, with pictures of the evidence and related preservation protocols.

## 3. Insect Development

Most frequently, forensic entomologists estimate the age of immature insects collected on a death scene and use this information as the minimum PMI [12]. The reference developmental data for the species that was collected on a death scene is necessary for such estimation. Because developmental data may vary between geographical populations of insects, it is recommended to use reference data from the closest population [10,13,14,15]. Although there is constant progress in this field, with new species and populations gaining developmental data, still much research needs to be done. A review of developmental datasets available for insect species colonizing cadavers in central Europe reveals that among the most extensively researched are cosmopolitan species that colonize cadavers shortly after death and that were frequently reported from indoor cases (Table 2). Although several important species have many datasets (e.g., *Lucilia sericata* or *Calliphora vicina*), there are still species that regularly breed in cadavers but for which no dataset has been published (e.g., *Lucilia caesar, Hydrotaea ignava* or *Necrobia violacea*) or only single datasets are available (e.g., *Stearibia nigriceps*, *Necrodes littoralis*, *Omosita colon* or *Necrobia rufipes*). These species should become the hot taxa for forensic entomology research in Europe. The understudied species ought to be identified also for other geographical regions.

Another point that needs our attention is the lack of standards and guidelines for developmental studies in forensic entomology. Several elements of the protocol for such studies were found to affect the quality of the resultant developmental data [16,17,18,19]. In addition, there is unnecessary variation in the type of development data provided and the way they are presented in publications. Standard research protocols emerge in mature sciences and I feel it is time to start this discussion in forensic entomology.

**Table 2 insects-12-00314-t002:** Developmental datasets available for the species that breed in large vertebrate cadavers in central Europe (species list compiled based on [20,21,22,23,24,25]).

Family	Species	Number of Published Datasets	Country of a Population’s Origin	References
Calliphoridae	*Calliphora vicina*	18	US,AT,GB,RU,CA,DE,IT,EG	[26,27,28,29,30,31,32,33,34,35,36,37,38,39,40,41,42,43]
	*Calliphora vomitoria*	7	US,GB,RU,DE	[26,29,31,33,37,44,45]
	*Chrysomya albiceps*	9	BR,RU,AT,ZA,CO,IR,EG	[14,26,39,46,47,48,49,50,51]
	*Lucilia caesar*	-	-	-
	*Lucilia sericata*	27	US,FI,IT,GB,RU,CA,AT,CO,IR,EG,TR,FR,EC,KR,CN	[26,27,28,30,31,33,36,48,52,53,54,55,56,57,58,59,60,61,62,63,64,65,66,67,68,69,70]
	*Phormia regina*	7	US,RU,CA,MX	[26,27,28,31,71,72,73]
	*Protophormia terraenovae*	7	US,GB,RU,AT,CA	[26,31,33,44,74,75,76]
Sarcophagidae	*Sarcophaga argyrostoma*	3	AT,DE,TR	[29,77,78]
	*Sarcophaga caerulescens*	-	-	-
Muscidae	*Hydrotaea dentipes*	-	-	-
	*Hydrotaea ignava*	-	-	-
	*Hydrotaea pilipes*	-	-	-
Fanniidae	*Fannia canicularis*	2	US,PL	[79,80]
	*Fannia scalaris*	-	-	-
	*Fannia leucosticta*	-	-	-
Piophilidae	*Stearibia nigriceps*	1	RU	[26]
Silphidae	*Necrodes littoralis*	2	PL	[81,82]
	*Thanatophilus rugosus*	1	DZ	[83]
	*Thanatophilus sinuatus*	1	CZ	[84]
Histeridae	*Margarinotus brunneus*	-	-	-
	*Saprinus planiusculus*	-	-	-
	*Saprinus semistriatus*	-	-	-
Staphylinidae	*Aleochara curtula*	-	-	-
	*Creophilus maxillosus*	5	US,CN,PL	[85,86,87,88,89]
	*Philonthus politus*	-	-	-
Dermestidae	*Dermestes frischii*	3	GB,ES,IT	[90,91,92]
	*Dermestes lardarius*	2	GB	[93,94]
	*Dermestes murinus*	-	-	-
Nitidulidae	*Omosita colon*	1	CN	[95]
Cleridae	*Necrobia rufipes*	1	CN	[96]
	*Necrobia violacea*	-	-	-
Pteromalidae	*Nasonia vitripennis*	5	AT,US,AU,CN,BR	[97,98,99,100,101]

AT—Austria, AU—Australia, BR—Brazil, CA—Canada, CN—China, CO—Colombia, CZ—Czech Republic, DE—Germany, DZ—Algeria, EC—Ecuador, EG—Egypt, ES—Spain, FI—Finland, FR—France, GB—United Kingdom, IR—Iran, IT—Italy, KR—Republic of Korea, MX—Mexico, PL—Poland, RU—Russian Federation, TR—Turkey, US—United States and ZA—South Africa.

## 4. Insect Succession

There are several forensic reasons to study insect succession on cadavers. First, these studies yield inventories of carrion insects for habitats and geographical locations that form a starting point for any further research in forensic entomology. Such inventories were published for many habitats and locations around the world (recently reviewed in [102]), but there are still white spots on this map.

Second, succession studies provide reference data on the pre-appearance interval (PAI) and the presence interval (PI) of particular insect taxa. Such data are essential to use insects that colonize cadavers late in decomposition, as their PAI may be longer than the development interval, and to get meaningful PMI it may be necessary to combine insect age with the PAI [5,11,103]. PAI may also support estimates of maximum PMI when insect evidence is absent [104,105]. PAI may be estimated using the temperature models for PAI [106]. However, such models are available only for some taxa, and for several important taxa (e.g., blow flies) PAI may not be estimated using the temperature data [107,108]. In such cases, insect succession studies with animal cadavers (preferably large pigs [102]) yield the best PAI reference data (e.g., average seasonal PAIs). As for the PI, it has a more complex causal background than PAI, its predictions are inherently related with larger inaccuracy (Figure 1) and currently it may be approximated only based on the reference data from succession studies. Although in some habitats and locations robust PAI or PI datasets are available for many taxa, usually there is shortage of such data (Table 3). In particular, indoor habitats need more attention. Therefore, pig decomposition studies to yield PAI and PI data of forensically relevant insects should be one of the priority research areas in forensic entomology.

Third, decomposition experiments using pig cadavers may be useful to validate the PMI estimation protocols [109]. Such experiments are especially suitable as proof-of-concept studies or initial validation studies [102]. Although validation of new methods is a priority in forensic sciences [110], datasets on the performance of insect-based methods for PMI are very limited. Validation using pig cadavers (ultimately also human cadavers [102]) should be another primary research area in forensic entomology (Section 6 of this article).

There are guidelines for decomposition studies in forensic entomology [102,111,112,113,114,115]. Still, however, more standardization is necessary, particularly in terms of the sampling frequency, insect identifications and the presentation of the results (summarized in [116]). We need to remember that PAI and PI data for particular taxa are necessary when results of the study are to be used for the estimation of PMI. Therefore, the data for immature insects should be prioritized. When only a few cadavers were used, a daily occurrence matrix may be the best choice to present the results in a forensically useful way [117,118]. When more cadavers were investigated, it may be necessary to present insect occurrences in a synthetic way, but still raw data from individual cadavers (or seasonal averages) should be given on the PAI and PI of particular insect taxa (e.g., [119]).

**Table 3 insects-12-00314-t003:** Datasets on the pre-appearance interval (PAI) and the presence interval (PI) of the species that breed in large vertebrate cadavers in central Europe (species list compiled based on [20,21,22,23,24,25]). I reviewed datasets derived from experiments performed in Europe and on pig cadavers only.

Family	Species	PAI		PI—Seasonal Data (Country/Habitat/Season/Stage)	References
		Temperature Model	Seasonal Data (Country/Habitat/Season/Stage)		
Calliphoridae	*Calliphora vicina*	-	PL/F/S/A,L1 IT/Ou/u/Au,W/A AT/Ou/u/S,Su/A PT/Ou/u/S,Su,Au,W/A PT/Ou/u/S,Au,W/O,L1,P ES/I/S,Su,Au,W/O	PL/F/S/A,L IT/Ou/u/Au,W/A AT/Ou/u/S,Su/A PT/Ou/u/S,Su,Au,W/A PT/Ou/u/S,Au,W/E,L,P ES/I/S,Su,Au,W/E	[22,119,120,121,122]
	*Calliphora vomitoria*	-	PL/F/S,Su,Au/A,L1,L3 IT/Ou/u/Au,W/A AT/Ou/u/S/A PT/Ou/u/S,Su,Au,W/A PT/Ou/u/S,W/O,L1,P ES/I/S/O	PL/F/S,Su,Au/A,L IT/Ou/u/Au,W/A AT/Ou/u/S/A PT/Ou/u/S,Su,Au,W/A PT/Ou/u/S,W/E,L,P ES/I/S/E	[22,25,119,120,121,122,123]
	*Chrysomya albiceps*	-	IT/Ou/u/Su,Au/A AT/Ou/u/Su/A PT/Ou/u/Su,Au/A PT/Ou/u/Su,Au/O,L1,P ES/I/S,Su,Au/O	IT/Ou/u/Su,Au/A AT/Ou/u/Su/A PT/Ou/u/Su,Au/A PT/Ou/u/Su,Au/E,L,P ES/I/S,Su,Au/E	[22,120,121,122]
	*Lucilia caesar*	-	PL/Ou/r/S,Su/A,L3 PL/F/S,Su,Au/A,L1,L3 IT/Ou/u/Su,Au,W/A PT/Ou/u/S,Su,Au/A PT/Ou/u/S,Su,Au/O,L1,P	PL/F/S,Su,Au/A,L IT/Ou/u/Su,Au,W/A PT/Ou/u/S,Su,Au/A PT/Ou/u/S,Su,Au/E,L,P	[21,25,119,120,121,123]
	*Lucilia sericata*	-	PL/Ou/r/S,Su/A IT/Ou/u/Su,Au,W/A PT/Ou/u/S,Su,Au/A PT/Ou/u/Au/O,L1,P ES/I/S,Su,Au/O	IT/Ou/u/Su,Au,W/A PT/Ou/u/S,Su,Au/A PT/Ou/u/Au/E,L,P ES/I/S,Su,Au/E	[21,120,121,122]
	*Phormia regina*	A	PL/Ou/r/S,Su/A PL/F/S,Su,Au/A,L1 AT/Ou/u/S,Su/A PL/F/S,Su/L3	PL/F/S,Su,Au/A,L AT/Ou/u/S,Su/A	[21,22,25,107,119,123]
	*Protophormia terraenovae*	-	AT/Ou/u/S,Su/A	AT/Ou/u/S,Su/A	[22]
Sarcophagidae	*Sarcophaga argyrostoma*	-	-	-	-
	*Sarcophaga caerulescens*	-	-	-	-
Muscidae	*Hydrotaea dentipes*	A	PL/F/S,Su,Au/A PT/Ou/u/S/A PL/F/S,Su/L1	PL/F/S,Su,Au/A PT/Ou/u/S/A PL/F/S,Su/L	[25,107,119,121,123]
	*Hydrotaea ignava*	A	PL/Ou/r/S,Su/A,L3 PL/F/S,Su,Au/A PT/Ou/u/S,Su,Au/A PL/F/S,Su/L1 PL/F/Su/L3	PL/F/S,Su,Au/A PT/Ou/u/S,Su,Au/A PL/F/S,Su/L	[21,25,107,119,121]
	*Hydrotaea pilipes*	-	PL/Ou/r/S,Su/A PL/F/S,Su,Au/A	PL/F/Su,Au/A	[21,25,123]
Fanniidae	*Fannia canicularis*	-	IT/Ou/u/Au,W/A	IT/Ou/u/Au,W/A	[120]
	*Fannia scalaris*	-	-	-	-
	*Fannia leucosticta*	-	-	-	-
Piophilidae	*Stearibia nigriceps*	A,O	PL/Ou/r/S,Su/A,L3 PL/F/S,Su,Au/A,L1 IT/Ou/u/Au/A PT/Ou/u/S,Su,Au/A PT/Ou/u/Su/E PT/Ou/u/S,Su,Au/L1,P PL/F/S,Su/L3	PL/F/S,Su,Au/A,L IT/Ou/u/Au/A PT/Ou/u/S,Su,Au/A PT/Ou/u/Su/E PT/Ou/u/S,Su,Au/L,P	[21,25,107,119,120,121,123]
Silphidae	*Necrodes littoralis*	A,L1	PL/Ou/r/S,Su/A,L1 PL/F/S,Su,Au/A,L1	PL/F/S,Su,Au/A,L	[21,25,103,119,123,124]
	*Thanatophilus rugosus*	-	PL/F/S,Su,Au/A IT/F/W/A PL/Ou/r/S/L3	PL/F/Su,Au/A IT/F/W/A PL/Ou/r/S/L3	[25,120,123,125]
	*Thanatophilus sinuatus*	A,L1	PL/Ou/r/S,Su/a PL/F/S,Su,Au/A IT/Ou/u/W/A PT/Ou/u/S,Au,W/A PL/Ou/r/S/L3	PL/F/S,Su,Au/A IT/Ou/u/W/A PT/Ou/u/S,Au,W/A PL/Ou/r/S/L3	[21,25,119,120,124,125,126]
Histeridae	*Margarinotus brunneus*	A	PL/Ou/r/S,Su/A PL/F/S,Su/A PT/Ou/u/S,Su,Au,W/A	PL/F/S,Su/A PT/Ou/u/S,Su,Au,W/A	[21,25,119,124,126]
	*Saprinus planiusculus*	A	PL/F/S/A	PL/F/S/A	[119,124]
	*Saprinus semistriatus*	A	PL/Ou/r/S,Su/A PL/F/S,Su,Au/A	PL/F/S,Su,Au/A	[21,25,119,123,124]
Staphylinidae	*Aleochara curtula*	-	PL/F/S,Su/A IT/Ou/u/W/A	IT/Ou/u/W/A	[25,120]
	*Creophilus maxillosus*	A,L1	PL/Ou/r/S,Su/A,L1 PL/F/S,Su,Au/A,L1 IT/Ou/u/Au,W/A PT/Ou/u/S,Su,Au,W/A	PL/F/S,Su,Au/A,L IT/Ou/u/Au,W/A PT/Ou/u/S,Su,Au,W/A	[21,25,119,120,123,124,126,127]
	*Philonthus politus*	A	PL/F/S,Su,Au/A IT/Ou/u/Au/A	IT/Ou/u/Au/A	[25,120,124]
Dermestidae	*Dermestes frischii*	-	PL/Ou/r/S,Su/A,L1 PT/Ou/u/S,Su,Au/A	PT/Ou/u/S,Su,Au/A	[21,126]
	*Dermestes lardarius*	-	-	-	-
	*Dermestes murinus*	-	PL/F/S,Su,Au/A PL/F/S/Lm	PL/F/Su,Au/A	[25,123]
Nitidulidae	*Omosita colon*	-	-	-	-
Cleridae	*Necrobia rufipes*	A	IT/Ou/u/Su,W/A PT/Ou/u/Su,Au/A	IT/Ou/u/Su,W/A PT/Ou/u/Su,Au/A	[120,124,126]
	*Necrobia violacea*	A	PL/Ou/r/S,Su/A PL/F/S,Su/A PT/Ou/u/S,Su,Au,W/A PL/F/S/L3	PT/Ou/u/S,Su,Au,W/A PL/F/S/L3	[21,25,119,124,126]
Pteromalidae	*Nasonia vitripennis*	-	AT/Ou/u/S/A	AT/Ou/u/S/A	[22]

A—adult stage PAI or PI, O—oviposition PAI, E—egg PI, L1—first instar larvae PAI, L—larval PI, L3—third instar larvae PAI or PI, Lm—mature larvae PAI or PI, P—puparial/pupal PAI or PI. S—spring, Su—summer, Au—autumn, W—winter. I—indoor habitats, Ou/u—outdoor, urban habitats, Ou/r—outdoor, rural habitats, F—forests. AT—Austria, ES—Spain, IT—Italy, PL—Poland, and PT—Portugal.

## 5. Temperature Conditions

The succession and development of insects on cadavers is largely dependent on the temperature [124,128,129]. When estimating PMI from insect succession or development, it is necessary to reconstruct temperature conditions. The accuracy of the PMI estimation depends largely on the accuracy of the reconstructed temperature conditions. This source of error is one of the most important.

Forensic entomologists frequently use temperature data from the local weather stations. Weather station temperatures can be corrected to adjust them to the peculiarities of a death scene [130,131,132,133,134]. Such corrections are based on the regression analysis between recordings made on a death scene and recordings from the station and for this reason they may be unfeasible [135]. Moreover, some authors indicate that the correction protocol has uncertain benefits for the accuracy of PMI estimation [135,136]. From the other side, there are robust experimental data indicating that the protocol improves the death scene temperatures [130,132,133,134]. It was found beneficial in casework, as well [5,132], although it was used infrequently [134]. The protocol may be impractical and its use may have a minor risk of deteriorating the death scene temperatures; however, it is the best tool we have and we should try to use it more frequently, particularly on outdoor death scenes. We need to remember that the protocol makes the weather station temperatures closer to the cadaver’s ambient temperatures only. Therefore, the corrected temperatures may still be far from the true temperatures experienced by the insects, because the protocol accounts for peculiarities of a death scene in terms of the factors that affect ambient air temperature only. In order to take into account other important factors a different approach is needed.

Some authors modelled temperature conditions in parked vehicles [137], containers [138] or specific urban and semi-natural habitats (e.g., cellars, attics or trailers) [139]. A model was also derived to extract heat profiles representing the temperatures experienced by insect populations growing on cadavers [140]. Charabidze and Hedouin [135] developed an algorithm to correct temperatures through a qualitative analysis of thermal-specific aspects of the case. The analysis consisted of six stages, starting from the conditions on a cadaver and moving towards the outside of the body. This research area is growing and both quantitative and qualitative approaches may be useful here.

The last factor that needs much more of our attention is the insect-driven thermogenesis. It has been discovered and extensively studied in aggregations of blow fly larvae [141,142,143,144,145,146,147,148,149,150,151,152]. Recently, insect-driven thermogenesis has been also reported for carrion beetles *Necrodes littoralis* L. (Silphidae), with evidence that heat is produced within the feeding matrix, which is formed by adult and larval beetles through spreading their exudates over the cadaver surface [153]. Thermogenesis in larval aggregations may be more common among carrion insects, and because it may substantially shorten the development interval, it should be factored when reconstructing temperature conditions [151,153].

When large aggregations of fly larvae (with elevated temperature) are present on a cadaver, Charabidze and Hedouin [135] suggest to use the minimum development time for the feeding stage of each species. Unfortunately, minimum times needed to reach the post-feeding phase in large aggregations of larvae are not available for any species. Accordingly, it may be tempting to use minimum development times from the laboratory development studies, as there are many such datasets (Table 2). However, in such studies minimum development times are recorded at high and constant temperatures that may be suboptimal for the insects. Experiments using the tracking of blow fly larvae within aggregations indicated that they have a strong preference for the hottest part of the aggregation [154]. This finding prompted the authors to state that the maximum temperatures of the aggregation represent the actual temperatures experienced by the larvae [154]. More recent data demonstrated that larvae continuously move between the periphery and the inside of the aggregation, with individual larvae spending from 16 to 68% (mean 43%) of their time at the aggregation periphery [155]. The periphery has a lower temperature than the inside of the aggregation [154]. For this reason, the heat gain of individual larvae may be smaller than if they were feeding for the whole time in the hottest part of the aggregation. The maximum temperature of the aggregation probably overestimates the true temperatures experienced by the larvae. Perhaps the heat benefit of the aggregated larvae is somehow related to the temperatures selected by the larvae along a thermal gradient [143,154,156]. Further research using tracking techniques to monitor heat benefits and the development time of individual larvae within large aggregations will be necessary to find the minimum development times or the optimal temperatures for the larvae that develop in an aggregation.

## 6. Challenging Evidence

All insect evidence can be challenging in some cases, and some types are always challenging. The puparia of flies and pupae of beetles are informative pieces of insect evidence, particularly on decomposed cadavers [5,11]. However, they are difficult to identify and it is difficult to estimate their age.

The identification of insect evidence is a necessary first step in any analysis. Significant progress has been recently made in this field, with several forensically important fly taxa gaining excellent identification keys for adult insects [13,157,158,159,160,161] and larvae [162,163,164]. Puparia should be the next step. Although some groups of carrion flies have useful descriptions of puparia [165], there is no forensically useful key for this type of insect evidence in any family of flies. This area is much less developed in the case of the forensically important beetles. There is just a single identification key for the larvae of beetles that colonize cadavers [166] and a single key for the adult carrion beetles (Silphidae) that frequent cadavers [167]. Although some descriptions of larval identification features have been published for forensically important species [168,169], this group needs more attention. Otherwise, we will still have to base our identifications on the taxonomic references that may be inaccessible to forensic entomologists with no experience in beetle taxonomy.

It is difficult to estimate the age of the fixed puparia of flies and pupae of beetles. The most promising techniques for aging such evidence consist of the qualitative morphological analyses of the intra-puparial forms of the flies [30,35,36]. Intra-puparial development has been documented for many forensically important species [35,65,78,170,171,172,173,174,175]. Although these techniques have obvious advantages (e.g., they cover most of the intra-puparial development, they are generally non-destructive, low-cost and they need a stereomicroscope only), they have also important disadvantages (e.g., they are qualitative in nature and therefore less accurate and they are also impractical due to the need to have an expert knowledge in the intra-puparial morphology) [82]. Recently, a simple-to-use technique has been developed for aging pupae of the carrion beetle *Necrodes littoralis* by means of the quantification of the eye-background contrast, with very encouraging results of the initial validation [82]. Similar quantitative techniques should be developed in other forensically important insects.

Empty puparia (i.e., hardened outer shells that remain upon the completion of immature development of some flies) are frequently collected on cadavers with long PMI, and their examination may provide an estimate of the minimum PMI [176]. This type of insect evidence poses specific difficulties. When estimating minimum PMI based on the empty puparium, it is necessary to take the post-eclosion interval (PEI) into account. The interval starts when an adult fly emerges from the puparium and ends when the empty puparium is being collected. PEI may be longer than the minimum PMI estimated based on the puparium. Although techniques to estimate PEI are being developed [177,178,179], they are far from the implementation to forensic casework. In a recent PMI simulation study, seasonal patterns of changes in PMI following various PEIs were revealed for the empty puparia of two species of flies, demonstrating that the simulation studies may guide estimation of the minimum PMI based on such challenging evidence [176].

## 7. Validation of the PMI Estimation Protocols

Forensic entomologists developed several methods for the estimation of PMI based on insect development [26,41,180,181,182] or succession [103,105,115,117,118,128,183]. As there are contemporary reviews of these methods [1,184], in this article I focus only on their validation (Table 4). Validation of a protocol for the estimation of PMI is of key importance, as it may demonstrate that the protocol provides robust evidence when used in a forensic context. Validation studies may also provide PMI errors that could be used to present a PMI estimate as a meaningful interval. Sometimes entomologists provide point estimates for PMI (e.g., [185,186]). The inaccuracy of the PMI estimate that usually has many and diverse sources, but which is inherently related to every analysis of insect evidence, should be explicit in casework. This may be accomplished by providing ranges for PMI. Therefore, an interval estimate for PMI should be a standard way to present the results of the insect evidence analysis. If we knew robust errors, they could be used to transform any PMI estimate (a point or a range) into a highly informative interval that takes into account all sources of inaccuracy. The error of estimation is the difference between the estimated and true PMI, expressed as a percentage of the true or estimated PMI (hereafter error I and II). If such errors were calculated for a reliable sample of PMI estimations, i.e., a large sample of forensic case reports with known true PMI or a large sample of PMI estimations for experimentally used human cadavers, they might robustly approximate the accuracy of the PMI estimation protocol in a forensic context. I believe that such errors could also yield a truly informative evaluation of the uncertainty in PMI estimates in casework.

Most of the validation studies in forensic entomology were proof-of-assumptions or proof-of-concept studies (Table 4). Experiments fully validating the estimation protocols were rare. Only a few such datasets have been published; most used pig cadavers and were replicated moderately, at most. Human cadavers in anthropology research facilities (i.e., body farms) could be used more extensively for that purpose. The estimation of PMI for such cadavers using mock crime scenarios could provide robust validation data. This research design is surprisingly underutilised at body farms.

Similarly, validations using casework data were infrequent (Table 4). In order to use the casework data for the validation, a true PMI needs to be specified based on a confession or a witness statement about when the victim was last seen alive, or other non-insect evidence. Although the non-insect evidence only approximates the true PMI sensu stricto, this is the only way to use casework data for the validation. However, published case reports rarely provide information on the true PMI sensu largo. In order to calculate the errors of insect-based protocols for PMI, I analyzed relevant case reports where the PMI was estimated based on insect development (Table 5) and separately based on insect succession (Table 6). Due to the imperfections of the data used, resultant errors need to be treated with caution. They are only rough approximations of the true errors of the insect-based protocols for PMI.

**Table 4 insects-12-00314-t004:** Validation of the protocols for the estimation of PMI based on insect evidence.

Type of the Validation	Aims	Development-Based Protocols		Succession-Based Protocols	
		Number of Studies	References	Number of Studies	References
Proof-of-assumptions study ^1^	Testing validity of the assumptions that are at the root of the protocol	26	[14,17,18,19,27,29,30,34,41,55,56,57,61,144,145,176,187,188,189,190,191,192,193,194,195,196]	56	[20,21,24,25,103,107,113,119,122,124,127,128,197,198,199,200,201,202,203,204,205,206,207,208,209,210,211,212,213,214,215,216,217,218,219,220,221,222,223,224,225,226,227,228,229,230,231,232,233,234,235,236,237,238,239,240]
Proof-of-concept study ^1^	Testing validity of the protocol as used in a simplified setting	12	[30,73,81,82,87,88,89,241,242,243,244,245]	3	[103,127,128]
Experimental validation with non-human cadavers	Testing validity of the protocol as used for non-human cadavers in an experimental setting	6	[109,241,246,247,248,249]	6	[105,106,183,249,250,251]
Experimental validation using human cadavers	Testing validity of the protocol as used for human cadavers in an experimental setting	0		1	[183]
Validation using casework data ^1^	Testing validity of the protocol as used in forensic casework	7	[46,118,182,185,252,253,254]	6	[5,108,118,186,255,256]

^1^ Selected studies were referenced.

There were surprisingly large differences between the cases. Errors I (differences between the true and estimated PMI expressed as the percentage of the true PMI) ranged from 0 to 83% for the development-based estimates (Table 5) and from 2 to 43% for the succession-based estimates (Table 6). Surprisingly, average errors were larger for the development-based estimates (22.3%) than the succession-based estimates (13.4%). Although the average difference between the true and estimated PMI was almost four times lower for the development-based estimation than the succession-based estimation (1.5 and 5.6 days respectively, Table 5 and Table 6), the latter type of the estimation was usually used for cadavers with a much larger PMI (Table 6); therefore, it had lower errors, which are relative values. Differences between the true and estimated PMI increased with the increase in the true PMI, and this relationship was particularly apparent when plotted for the development-based estimates (Figure 2).

**Table 5 insects-12-00314-t005:** Errors of the protocols for the estimation of PMI based on insect development.

Reference	N	True PMI ^1^ (days)		Difference True-Estimated PMI ^2^ (days)		Error I ^3^ (%)		Error II ^4^ (%)		Remarks
		Mean	Range	Mean	Range	Mean	Range	Mean	Range	
Goff et al., 1988 [252]	2	5.5	5–6	0.375	0.25–0.5	6.65	5–8.3	6.95	4.8–9.1	-
Kashyap, Pillay, 1989 [185]	16	4.9	0.5–9	0.438	0–1	13.74	0–50	11.65	0–33.3	No mention of temperature data
Grassberger et al., 2003 [46]	1	17	-	3	-	17.65	-	20	-	-
Reibe et al., 2010 [182]	1	4	-	0.125	-	3.125	-	3.03	-	-
Pohjoismäki et al., 2010 [253]	7	10.6	5–19	4.57	2.5–7	48.96	21.1–83.3	144.2	26.7–500	Single average temperature assumed in all cases (24 °C)
Bugelli et al., 2015 [254]	4	4.0	2–6	0.94	0.5–1.5	23.75	20–25	31.25	25–33	-

N—a number of PMI estimations in a dataset. ^1^ PMI determined based on non-insect evidence (a confession, a witness statement about when the victim was last seen alive, etc.). ^2^ An absolute difference between the true PMI and the PMI estimated based on insect development. When the estimated PMI was presented as an interval, I calculated absolute differences between the true PMI and the lower and upper limit of the estimated interval and then averaged them to get the difference between the true and estimated PMI. ^3^ Error I = (the difference between the true and estimated PMI/true PMI) × 100. ^4^ Error II = (the difference between the true and estimated PMI/estimated PMI) × 100. When the estimated PMI was presented as an interval, a midpoint of the interval was used in denominator.

**Table 6 insects-12-00314-t006:** Errors of the protocols for the estimation of PMI based on insect succession.

Reference	N	True PMI ^1^ (days)	Estimated PMI (days)	Difference True-Estimated PMI ^2^ (days)	Error I ^3^ (%)	Error II ^4^ (%)	Remarks
Goff et al., 1986 [256]	1	20	19–20	0.5	2.5	2.6	-
Goff and Odom, 1987 [186]	1	53	≥52	1	1.9	1.9	-
Goff and Flynn, 1991 [255]	1	38	34–39	2.5	6.6	6.8	-
Schoenly et al., 1996 [118]	2	11	10.5–11	0.25	2.3	2.3	-
		36	34–36	1	2.8	2.9	
Archer, 2014 [108]	1	21	16–34	9	42.9	36	-
Matuszewski and Mądra-Bielewicz, 2019 [5]	1	72	30–64	25	34.7	53.2	Less reliable true PMI

N—a number of PMI estimations in a dataset. ^1^ PMI determined based on non-insect evidence (a confession, a witness statement about when the victim was last seen alive, etc.). ^2^ An absolute difference between the true PMI and the PMI estimated based on insect succession. When the estimated PMI was presented as an interval, I calculated absolute differences between the true PMI and the lower and upper limit of the estimated interval and then averaged them to get the difference between the true and estimated PMI. ^3^ Error I = (the difference between the true and estimated PMI/true PMI) × 100. ^4^ Error II = (the difference between true and estimated PMI/estimated PMI) × 100. When the estimated PMI was presented as an interval, a midpoint of the interval was used in denominator.

Summarizing, the protocols for the estimation of PMI based on insect evidence usually lack errors and their validity has been rather poorly demonstrated in a true forensic context. Therefore, validation studies using pig or human cadavers and casework data should be prioritized in forensic entomology. I think this is our greatest challenge.

## 8. Conclusions

Although the set of challenges elucidated in this article is somehow subjective, I believe that most forensic entomologists would construct similar sets. Some challenges should focus more of our attention, with priority for the resultant research. This applies, in particular, to the validation research, as well as to development and succession research. Studies on thermogenesis in larval aggregations on cadavers should be prioritized as well. There are also highly important challenges of educational and promotional nature. Although we should look for more optimal guidelines for insect sampling on a death scene, and this is a scientific task, improvement in the samples taken by a law enforcement personnel depends equally or even more on the promotion of forensic entomology among its end-users and on the education of the officers or medical examiners that collect insect evidence on death scenes.

## Figures and Tables

**Figure 1 insects-12-00314-f001:**
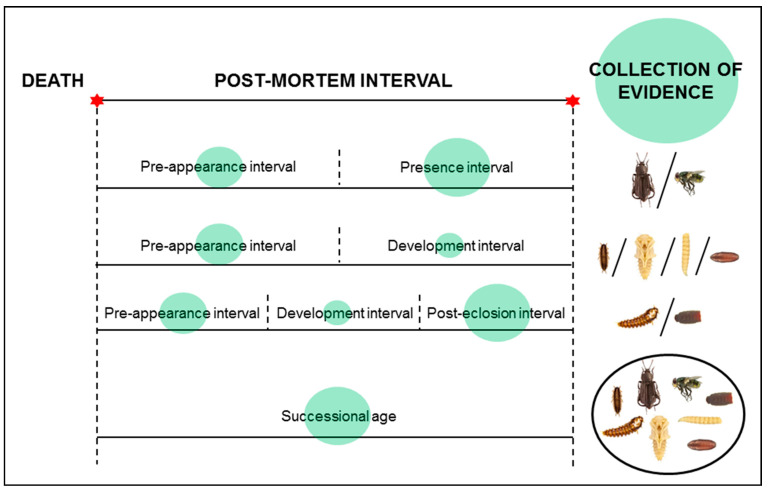
Sources of inaccuracy in the estimation of the post-mortem interval (PMI) based on insect evidence. Green circles represent the sources, their size represents importance of the sources. Pictures of insects were made by Anna Mądra-Bielewicz (Poznań, Poland).

**Figure 2 insects-12-00314-f002:**
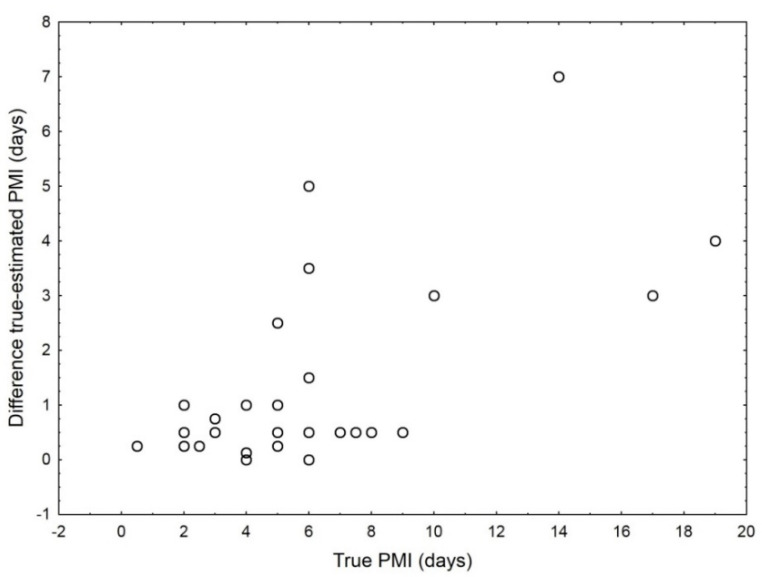
Differences between the true and estimated PMI plotted against the true PMI for the estimations based on insect development that were referenced in Table 5.

**Table 1 insects-12-00314-t001:** A sketch of guidelines for the collection of insect evidence on a death scene. To make them useful to non-entomologists, they should be combined with pictures of insect evidence and protocols suitable for the preservation of particular pieces of evidence.

State of a Cadaver	Insect Evidence	
	What to Look For	Where to Look For
Relatively fresh	Eggs or larvae of flies	Natural orifices (particularly of the head), wounds
Signs of putrefaction (bloating, marbling, etc.)	Larvae of flies	Natural orifices, wounds, interface cadaver/ground
Signs of active decay (large masses of insect larvae, stench of decay, leakage of decomposition fluids, etc.)	1. Larvae (particularly post-feeding) of flies 2. Larvae of beetles	1. Larval masses, the surface of soil (outdoor scenarios) or the floor (indoor scenarios) in the vicinity of a cadaver 2. Larval masses, clothes and cadaver surface, the soil in the vicinity of a cadaver (outdoor scenarios, soil samples are recommended), the floor in the vicinity of a cadaver (indoor scenarios)
Signs of advanced decay (exposed bones; greasy by-products of active decay, darkening of the remaining skin, etc.)	1. Puparia (full and empty) of flies 2. Larvae and pupae of beetles 3. Larvae of late-colonizing flies (e.g., skipper flies)	1. The soil in the vicinity of a cadaver (outdoor scenarios, soil samples are recommended), the floor (under carpets or furniture) in the vicinity of a cadaver (indoor scenarios), pockets and foldings of clothes, cadaver surface (all scenarios) 2. Larval masses, clothes and cadaver surface, the soil in the vicinity of a cadaver (outdoor scenarios, soil samples are recommended), the floor in the vicinity of a cadaver (indoor scenarios) 3. Larval masses, the surface of soil (outdoor scenarios) or the floor (indoor scenarios) in the vicinity of a cadaver
Signs of minimal insect infestation (e.g., massive putrefaction or mummification)	All types of insect evidence	Natural orifices, wounds, clothes and cadaver surface, the soil (outdoor scenarios) or the floor (indoor scenarios) in the vicinity of a cadaver

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
