# Peer review of "Post-Mortem Interval Estimation Based on Insect Evidence: Current Challenges"

_insects, 2021, doi:10.3390/insects12040314_

Round 1

Reviewer 1 Report

The manuscript presents a comprehensive overview and includes an extensive collection of relevant literature. Research ideas and suggestions for improvement are given in each chapter. Chapter 7 "PMI estimation protocols" should be revised.

For details please see additional file.

Author Response

1. Introduction:
- line 20: add carrion to the first sentence.

Added. 

It should be mentioned that PMI estimations based on insects could only indicate minimal intervals.

I disagree. Methods based on insect succession usually provide both minimum and maximum PMI estimate. Methods based on development when they are coupled with PAI provide also minimum and maximum PMI. Only when PMI is based solely on the age of immature insects this is the minimum PMI, however in some cases this may be inaccurate, for instance when PMI is estimated based on the insects that colonized the body before death (i.e. myiasis). Therefore, it would be too narrow and inaccurate to limit in the introduction the scope of PMI estimation just to the minimum PMI. If necessary, I limited my discussion just to the minimum PMI, for instance when I discussed estimation based on an empty puparium, but to say that all PMI estimations can only indicate the minimum PMI is simply untrue.

Occurrences such as prior freezing of a body might later be undetectable. The pre-appearance interval is mentioned later in the manuscript and should be addressed in the introduction.

Actually, PAI is addressed in this section, in figure 1. I would prefer not to expand the introduction. The manuscript already has 23 journal pages and I believe it is not necessary to discuss more extensively PAI and other related concepts in the introduction.

2. Collection of insect evidence
- line 46: [5] is indicated as comparison of samples taken by non-experts and experts. Unfortunately, this comparison is not apparent in the reference.

This was not the purpose of that paper, I agree this is not apparent there. But Tables 1 and 2 of this manuscript and some unpublished data enabled this comparison in the current paper. I reworded this portion of the manuscript to avoid any confusion that the comparison was already made in the previous paper (see first paragraph of section 2). 

line 62: According to the author non-experts samples were not diverse enough in [5], here it is recommended for non-experts to sample less. “However, to estimate PMI only a small part of it is necessary.” How can these two statements coexist?

Non-expert samples were less diverse and did not contain insect evidence, based on which PMI has finally been estimated in that case.  Therefore, non-experts sampled wrong insect evidence, and this was an error. If they knew what and where to sample, the sample could be similarly small but more informative, as it would contain better insect evidence. I feel that this message is clear in section 2. 
- line 73: … what insect evidence is …

I reworded this sentence.

- table 1: guidelines for sampling according to state of cadaver and most developmentally advanced insects is dangerous: state of cadaver might be ambiguous and the most developmentally advanced individuals might not be easily recognized. For non-specialists it might be more advisable to collect too much than to limit collection! Sampling “most successionally advanced” individuals is specialist knowledge!

Table 1 is just a draft for discussion. To use guidelines included in this table, one needs no knowledge of insect sucession or development. The knowledge is necessary to construct or reshape guidelines, but not to use them. A non-expert needs only to check what insects should he look for and where, and in this framework only broad entomological categories are used, e.g. larvae of flies, puparia of flies etc. To make the table fully usable it should be combined with pictures of insect evidence, but I feel the table is a good starting point for further work in this field. I also had some reservations to use state of the cadaver here, that's why the table is "a sketch of guidelines". However, state of the cadaver is the best way to focus evidence sampling on a death scene, and the thing for discussion is how to use it for this purpose.  

3. Insect development
- very important points! Unfortunately, manuscripts containing developmental data for a specific species are often rejected due to lack of originality. Publication of developmental data based on established guidelines must be facilitated.

I fully agree.

4. Insect succession
Reasons to study insect succession: 1. Inventories, 2. Reference for pre-appearance interval (PAI), 3. Validate PMI estimation protocols
- table 3 is mentioned several times in a short section of text (lines 118-124), please reduce.

Reduced.

6. Challenging evidence
There are more challenges to insect evidence than interpreting puparia, e.g. very long and/or wrong storage, aging of adult insects etc. The chapter should be expanded.

I agree. However, this section was originally considered to be the review of the most important challenges. Therefore, I would not like to expand this part of the paper. The article already has 23 journal pages and it is the longest paper I have ever written. Besides, aging of adult insects (and related challenges) were recently reviewed by Amendt et al., 2021, Time flies..., Diagnostics... 

7. PMI estimation protocols
- line 252 Even though this chapter is titled “PMI estimation protocols”, it contains no protocols but discusses validation studies. The title should reflect this.

Good point, thank you. I changed the title.

- table 4 is mentioned four times within two consecutive paragraphs (lines 275-284), please reduce

Reduced.

- line 319: this paragraph calculates error rates from published case reports. First of all, most reports contain cases in which methods and estimations worked out correctly. They are predominantly used to show successes. Very few researchers will publish “failed” casework and expertise. This is shown in the fact, that the cases used in the manuscript to illustrate high error rates do not seem so bad on
closer examination. Bugelli et al. 2015 present 8 cases. The author of this manuscript calculated a range for the error in these cases to be 20-100%.
ï‚· Case 1: mPMI was estimated to be 4-5 days; “other evidence” (not specified in publication) indicated 5-6 days.
ï‚· Case 2: mPMI estimated 3 days; person last seen alive 3 days before discovery.
ï‚· Case 3: mPMI estimated 3-5 days; person last seen alive 5 days before discovery.
ï‚· Case 4: mPMI 21 days; person last seen alive 21 days before discovery.
ï‚· Case 5: mPMI 2 days; person last seen alive 48 h before discovery.
ï‚· Case 6: mPMI 2 days or 6-8 days; person last seen alive 2 days before discovery, myasis diagnosed 5 days before (discovery?).
ï‚· Case 7: no non-insect evidence to corroborate PMI
ï‚· Case 8: mPMI 1 day, person last seen alive 1 day before discovery.

I would like to thank this reviewer. Due to his comment I realized that Bugelli et al. provided contradictory data in their paper. Originally I used data from their Table 1, but after reading the above comment (with different data for some cases), I noticed that for some of Bugelli et al. cases contradictory data are provided in case descriptions and the table. Therefore, I removed from the ananlyses cases with such contradictions. Moreover, I had to recalculate errors for Bugelli paper (now only four cases weere included). I also had to update Figure 2 and text of this section. 

And when it comes to the substantive answer to the reviewer's comment, I insist that no casework were selected to ilustrate anything in the manuscript. I just took for the analyses all the published cases of which I was aware and which provided estimated and true PMI. I understand  reservations of this reviewer. However, in some of the cases insect-based methods yielded estimates, that were far from good, e.g. some Pohjoismaki cases had large error rates (50% or more of the true PMI). Similarly, Archer case for the succession estimates had about 43% error rate. Accordingly, assumption of the reviewer that only good estimates (successes) are published do not hold. And finally the average error rate for the development-based estimations was about 22%. If only accurate estimates were published this should be much lower. Personally, I think that this very rough analysis gave average error rates close to the true ones for these methods. 

The moment the person was last seen alive is not the true time of death, as the person reportedly was not dead by that time. It can therefore not be used as indicator for a true PMI, which must by definition be shorter. In fact, the estimation error cannot be calculated in these cases because true PMI is unknown. Pohjoismäki et al. 2010 present 9 cases, which are only sparsely described. For all cases a constant temperature of 24°C was assumed and a range of 18-30°C for an alternative estimation. I do not think that such data can be read as validation of casework, least used to calculate error rates.
Similar reservations are in order for table 6 (line 310). In Archer (2014) the estimated PMI range included the true PMI, while in Matuszewski et al. (2019) it did not. Error rates are calculated to be similar in both cases.

I agree. Time of the last seen alive is not a true PMI sensu stricto. This may be only some approximation of the true PMI. True PMI is usually shorter, so the resultant error rates will be larger. But only a little, usually the time last seen alive is close to the time of death. I also agree that some cases used in the analyses were very sparsely described. However, removing all these cases from the analyses would make them pointless. And they still provide worthwhile look at the accuracy of insect-based methods. When I decided to include this in this paper, I wanted to make this as a starting point for further analyses, the rough reference value for the methods (currently no such data exist) and the encouragement to publish good descriptions of case reports. I understand that the data is imperfect, but I feel these analyses were worthwhile (based on the comments of the reviewers, this section attracted their attention most). All the weaknesses of the data are explicit in Tables 5 and 6 and their footnotes, but to highlight them I added few sentences (third caption of section 6) to indicate a very initial nature of these analyses.   

Also, calculation of difference between true and estimated PMI is unclear. The possibility of being the true PMI is equal for each day in the time span estimated. The true PMI was within this range. How can a meaningful difference be calculated between true and estimated PMI?

When the estimated PMI was presented as an interval (some cases gave point estimates), I calculated absolute differences between the true PMI and lower and upper limit of the estimated interval and then averaged them to get the meaningful difference between true and estimated PMI. This definition was included in the footnote to Tables 5 and 6.

Please also explain the different meanings of error rates I and II.

They are explained in the footnote to Tables 5 and 6.

It would be helpful to include instructions on how to calculate and apply error rates in casework.

Usually you cannot calculate error rate in casework. The true PMI is to be estimated, it is usually unknown for the expert, so the error is also unknown. We may calculate the error rate of the method based on the estimations with known true PMI, but this is the general knowledge and not the specific knowledge for the given case. Can we use such general error rates in casework? I would say this is the best way to provide robust information about inaccuracy of  particular estimations. But this topic deserves a separate paper and I am not able to lay it shortly in this paper. 

Additionally, guidelines or ideas for designing useful validation studies would be immensely valuable.

I believe that the whole section 6 provides some guidelines/ideas for the validation studies. I am not able to provide more specific instructions, as it would necessitate large expansion of this section.

Reviewer 2 Report

The manuscript is well written and the author provides good advice for forensic entomologists. However, he did not address at least one very important change that is necessary if forensic entomology is to become a normal science.

Clearly an estimate of PMI (or related values) should be a range rather than a single value. Consider the most simple analysis, predicting the age of a larva based on size. For any set of environmental conditions, even if perfectly known, and for the same species and regional genotype, a given size will occur at more than one age.

Because the author does not think this way, he uses the wrong concept of error. If forensic entomologists estimated PMI correctly one would not refer to the estimate as being “close to the true PMI.” Instead, one would care whether the estimate “includes the true PMI.”  In statistics a standard measure of performance for a prediction method that generates a range (continuous or not) is the coverage (which combines accuracy and ability to reject incorrect values). Coverage is a relevant error rate for assessing PMI estimation method validity, not some single percentage value.

Figure 2 is very difficult to interpret because it is based on point estimates. Perhaps those predictions of PMI were 100% accurate.

The manuscript should include discussion of these issues.

Other comment

There is another important reason it is useful to know PAI. It can support an estimate of PMImax for an insect-free corpse.

Author Response

The manuscript is well written and the author provides good advice for forensic entomologists. However, he did not address at least one very important change that is necessary if forensic entomology is to become a normal science.

Clearly an estimate of PMI (or related values) should be a range rather than a single value. Consider the most simple analysis, predicting the age of a larva based on size. For any set of environmental conditions, even if perfectly known, and for the same species and regional genotype, a given size will occur at more than one age.

I agree. To make this explicit I added a short passage in section 6.

Because the author does not think this way, he uses the wrong concept of error. If forensic entomologists estimated PMI correctly one would not refer to the estimate as being “close to the true PMI.” Instead, one would care whether the estimate “includes the true PMI.”  In statistics a standard measure of performance for a prediction method that generates a range (continuous or not) is the coverage (which combines accuracy and ability to reject incorrect values). Coverage is a relevant error rate for assessing PMI estimation method validity, not some single percentage value.

I am not sure if I  understood this comment correctly. I don't know why this reviewer assumed that I support point estimates for PMI. I do not. More importantly, we have to consider "how close the true PMI” is to the estimated PMI (point or interval)? This is the essence of error. Coverage, if I unerstand it correctly is a very different measure and I am not sure if this may be termed a measure of error, as it says nothing about this distance. Coverage probability for the PMI range, I assume, is the probability that an interval cover the true PMI. When this interval will be very wide it will have almost 100% coverage, but it will be highly inaccurate. Error is related to the width of the interval, and how close the interval (its limits) is to the true PMI. Therefore, error rate needs to reflect the distance between true PMI and estimated PMI (point or interval). And that is why I used error rates in section 6 of my paper in the way defined in the footnotes to Tables 5 and 6. This is also the reason, why this section need not to include the coverage. The section is about errors.

Figure 2 is very difficult to interpret because it is based on point estimates. Perhaps those predictions of PMI were 100% accurate.

This figure is not based on point estimates, only some of the estimates used to get the difference between true and estimated PMI were point estimates, but other were interval estimates (see table 5 and its footnote for details). When I calculated error for the interval estimates, I took the absolute differences between the true PMI and lower and upper limit of the estimated interval and then averaged them to get the difference between true and estimated PMI. These definitions are in the footnotes to tables 5 and 6.

We cannot say that an estimate (interval) is 100% accurate when it contains the true PMI. When an interval is very wide, for instance it spans 0-2 years (sorry for this exaggeration), it will contain almost all true PMIs. But can we say that this interval is 100% accurate. No, it is in some ways 100% inaccurate, it simply says nothing useful about the true PMI.

The manuscript should include discussion of these issues.

Other comment

There is another important reason it is useful to know PAI. It can support an estimate of PMImax for an insect-free corpse.

Thank you, I added this in section 4.

Reviewer 3 Report

This is an excellent contribution, the author is to be congratulated on an important review of this subject, one that is overdue. I fully agree with the author's comment about the importance of the silphids as well. I think this should be published as is. 

Author Response

This is an excellent contribution, the author is to be congratulated on an important review of this subject, one that is overdue. I fully agree with the author's comment about the importance of the silphids as well. I think this should be published as is. 

Thank you.

Round 2

Reviewer 1 Report

Thank you for your detailed answers to my comments. It is true that the manuscript is rather long.

PIA: In figure 1 you acknowledge that the estimation or calculation of PAI is an important source of inaccuracy. Why it should be less important than the presence interval, however, is not clear to me. In chapter 4 you point out that there is a shortage of such data. Estimations of a maximal PMI based on forensic entomology alone therefore still seems very dangerous to me. However, as this does not seem to be the main point of your manuscript I of course accept your answer.

My comment did not have the intention to imply that you selected cases to illustrate anything, sorry that it was apparently misleading. The aim was to point out that results in Bugelli did not seem so bad to me. I am glad that my comments led to the identification of a misleading publication, which should be pointed out maybe in a future review of casework in forensic entomology. The method chosen by Pohjoismäki to analyze cases is more than questionable and surely does not reflect the intentions of all case report authors.

I agree that the analysis of the case reports is worthwhile and very interesting. However, an explanation for the meaning of error rates I and II is still missing. The footnotes of tables 5 and 6 explain how they were calculated, but what explanatory power of these two values have? What conclusions can be drawn from error rate I and are they different from error rate II conclusions? I know this would add even more text, but I am confident that the Information will be valuable for readers.

Author Response

PIA: In figure 1 you acknowledge that the estimation or calculation of PAI is an important source of inaccuracy. Why it should be less important than the presence interval, however, is not clear to me. In chapter 4 you point out that there is a shortage of such data. Estimations of a maximal PMI based on forensic entomology alone therefore still seems very dangerous to me. However, as this does not seem to be the main point of your manuscript I of course accept your answer.

Presence interval of adult insects has more complex causal background than PAI. In case of some taxa PAI is very stongly related to a single factor, a temperature preceding the appaerance. So, sometimes it may simply be estimated using temperature data. Presence interval is much more difficult to estimate. We have no methods for such estimation. The only thing we may do is to use seasonal (or monthly) averages from succession experiments. There is however the problem that usually these data will be highly inaccurate, as the presence interval largely depends on the season, actual temperatures, mass of the cadaver, other insects that colonized a cadaver etc. That is why the green circle in fig 1 is larger for PI than PAI. I added a sentence in section 4 to clarify this.

My comment did not have the intention to imply that you selected cases to illustrate anything, sorry that it was apparently misleading. The aim was to point out that results in Bugelli did not seem so bad to me. I am glad that my comments led to the identification of a misleading publication, which should be pointed out maybe in a future review of casework in forensic entomology. The method chosen by Pohjoismäki to analyze cases is more than questionable and surely does not reflect the intentions of all case report authors.

I agree, such a paper would be highly beneficial.

I agree that the analysis of the case reports is worthwhile and very interesting. However, an explanation for the meaning of error rates I and II is still missing. The footnotes of tables 5 and 6 explain how they were calculated, but what explanatory power of these two values have? What conclusions can be drawn from error rate I and are they different from error rate II conclusions? I know this would add even more text, but I am confident that the Information will be valuable for readers.

I added some clarifications in section 6.